# Role of export industries on ozone pollution and its precursors in China

Jiamin Ou[1,2,3,13], Zhijiong Huang [4,13], Zbigniew Klimont [3✉], Guanglin Jia[5], Shaohui Zhang [3,6], Cheng Li[7], Jing Meng [8], Zhifu Mi [8], Heran Zheng [2,9], Yuli Shan[10], Peter K. K. Louie[11], Junyu Zheng [4✉] & Dabo Guan [8,12✉]

This study seeks to estimate how global supply chain relocates emissions of tropospheric ozone precursors and its impacts in shaping ozone formation. Here we show that goods produced in China for foreign markets lead to an increase of domestic non-methane volatile organic compounds (NMVOCs) emissions by 3.5 million tons in 2013; about 13% of the national total or, equivalent to half of emissions from European Union. Production for export increases concentration of NMVOCs (including some carcinogenic species) and peak ozone levels by 20–30% and 6–15% respectively, in the coastal areas. It contributes to an estimated 16,889 (3,839–30,663, 95% CI) premature deaths annually combining the effects of NMVOCs and ozone, but could be reduced by nearly 40% by closing the technology gap between China and EU. Export demand also alters the emission ratios between NMVOCs and nitrogen oxides and hence the ozone chemistry in the east and south coast.

[1] Department of Sociology, Utrecht University, Utrecht 3584 CH, the Netherlands. [2] School of International Development, University of East Anglia, Norwich NR4 7JT, UK. [3] International Institute for Applied Systems Analysis, Schlossplatz 1, A-2361 Laxenburg, Austria. [4] Institute for Environmental and Climate Research, Jinan University, Guangzhou, China. [5] School of Environment and Energy, South China University of Technology, University Town, Guangzhou, China. [6] School of Economics and Management, Beihang University, 37 Xueyuan Road, 100091 Beijing, China. [7] Research Center for Eco-Environmental Engineering, Dongguan University of Technology, Dongguan, China. [8] The Bartlett School of Construction and Project Management, University College London, London WC1E 7HB, UK. [9] Industrial Ecology Programme, Norwegian University of Science and Technology, Trondheim, Norway. [10] Integrated Research on Energy, Environment and Society (IREES), Energy and Sustainability Research Institute Groningen, University of Groningen, Groningen 9747 AG, the Netherlands. [11] Hong Kong Environmental Protection Department, 5 Gloucester Road, Hong Kong, China. [12] Department of Earth System Science, Tsinghua University, 100084 Beijing, China. [13] These authors contributed equally: Jiamin Ou, Zhijiong Huang. ✉email: klimont@iiasa.ac.at; zheng.junyu@gmail.com; guandabo@hotmail.com

Ozone ($O_3$) in the troposphere is an important air pollutant detrimental to human health and ecosystem productivity[1]. In the past few decades, the entire Northern Hemisphere has seen significant increases in tropospheric $O_3$ pollution, especially in the East and South Asia[2]. Since China started to include $O_3$ to its national monitoring network in 2013, the recorded hourly $O_3$ increased by 16–27% from 2013 to 2017[3]. The $O_3$ exposure metrics (cumulative $O_3$ concentration) increased even more by 57–77%[3]. The present extent of $O_3$ pollution, in terms of the exposure of humans and vegetation, is greater in China than in any other developed region of the world with comprehensive $O_3$ monitoring[3].

Globally, there are continuous efforts to capture the dynamics of tropospheric $O_3$ pollution, and its causes and impacts. As a secondary pollutant, $O_3$ in the troposphere is not directly emitted by human activities. Rather, it is formed from precursor emissions of non-methane-volatile organic compounds (NMVOCs), nitrogen oxide (NOx), carbon monoxide (CO) and others under photochemical reactions[4]. Studies have shown that the spatial distribution of precursor emissions dominates global tropospheric $O_3$[5]. This can be attributed to the variations of photochemical reaction rates, convection and precursor sensitivities that affect the effectiveness of $O_3$ formation in different latitudes. The process of globalisation has connected countries better than ever and relocated a large amount of precursor emissions. More and more production activities have shifted from developed to developing countries. Among the later ones, China is undoubtedly the largest export economy[6]. Millions of tonnes of goods associated with $O_3$ precursor emissions are produced domestically and shipped and consumed elsewhere in the world.

Several studies have been conducted to investigate how the role of the world's factory has contributed to the domestic pollution and greenhouse gas emissions in China[7–10]. For example, Zhang et al.[7] studied how international trade has contributed to the global distribution of fine particulate matter ($PM_{2.5}$) pollution, and showed that around 10–20% of the premature mortality attributable to $PM_{2.5}$ in China was attributed to the demand of export. However, the impact of export on $O_3$ pollution in China is still largely unknown. Tropospheric $O_3$ distinguishes with $PM_{2.5}$ in terms of their precursors, formation regimes and sensitivities to other environmental factors such as sunlight and temperature. Indeed, there is a contrasting trend of $O_3$ and $PM_{2.5}$ in China. In contrast to the above-mentioned increase in $O_3$, $PM_{2.5}$ in eastern China has seen an annual decrease of around 7% from 2013 to 2017[11,12]. It suggests that knowledge and experiences in $PM_{2.5}$ are not necessarily applicable to $O_3$.

An understanding of the role of export industries in China's $O_3$ pollution might open up new opportunities to tackle the persistent growth of $O_3$ and its precursors in China. In addition to the rise of ambient $O_3$ levels in China, its precursor—NMVOCs—is also growing persistently in contrast to the sharp decrease in NOx and other primary pollutants[13,14]. The persistent growth of NMVOCs is mainly due to the increase in emissions from industrial processes and solvent use (+36%), while the NMVOCs from transport had decreased by 21% from 2010 to 2017[14]. In addition to the contribution of $O_3$ formation, some NMVOC species, such as benzene, toluene, ethylbenzene and xylenes (BTEX in short), have well-documented influences on the central nervous system and immune functions[15]. The debate about the priority of controlling NMVOCs or NOx to reduce $O_3$ in China led to policies asking for stronger reduction of NOx, delaying sector-wide NMVOC control[14,16]. However, there is agreement that controlling NMVOCs has not only its own merit (e.g., toxicity), but would help to alleviate the local/urban $O_3$ increases following NOx policy[4]. Among China's top export goods, many of them are associated with intensive NMVOC emissions,

including but not limited to vehicle parts, wood furniture, coke, integrated circuits, shoes and leather products. It is therefore important to understand the role of international export in China's $O_3$ formation and its precursors, and to explore new opportunities to curb the worrying growth of $O_3$ and NMVOCs in China.

Studies in China can also partly reveal how the global supply chain has shaped the $O_3$ formation in the low- and mid-latitudes of the Northern Hemisphere. Since 1980, a large proportion of $O_3$ precursor emissions have shifted from developed to developing regions. While the absolute change of $O_3$ in the world from 1980 to 2010 has been investigated[5], it is not clear how the emissions embodied in and relocated by the global supply chain have contributed to the present $O_3$ pollution, especially those in the exporting countries. Exporting countries might not only see an increase in the emissions of $O_3$ precursors, but a shift of the $O_3$ formation chemistry due to disproportionate changes of NMVOCs and NOx emissions[4,17]. As a vast country across a wide range of latitudes, the role of export industries in China has important implications for other countries.

Therefore, we present a consumption-based study on the tropospheric $O_3$ pollution in China with a focus on export. We utilise China's 2012 multiregional input–output (MRIO) table, the Global Trade Analysis Database (GTAP) and an air-quality model to estimate the contribution of export demand on $O_3$ precursors and its formation in China. Health burden associated with ambient $O_3$ and BTEX is estimated, followed by possible pathways to reduce the export footprint and aid the $O_3$ pollution control in China. Implications for other countries in southern Asia and Africa as the new receptors of China's export capacities are discussed.

## Results and discussion

**Scale of precursor emissions and change in chemistry.** The demand of export increases not only the production activities related to direct export products, but also the activities from power supply sectors, transportation and others to support the production of export goods. Nationally, export explains 13%, 15% and 10% of the NMVOCs, NOx and CO emissions in 2013, respectively. As the world's largest exporter for a lot of VOC-relevant products, the contribution from export industries to national NMVOC emissions was not as high as expected. However, it does not mean that we can downplay the role of export. Though being dwarfed by China's total emission budget, export-driven NMVOC emissions (3,794 kt in 2013) were equivalent to half the emissions of the European Union (EU, including EU-27 and UK)[18]. If such emissions were generated by a single country, it would be ranked as the 10th largest NMVOC emitter in the world[19].

The impact of export is highly uneven in China. Over 65% of China's astonishingly high export-related gross domestic product (GDP) comes from Guangdong, Fujian, Yangtze River Delta and Shandong (see Supplementary Fig. 1). As a result, the export-driven NMVOC emissions originate mainly from these regions. Around 18–26% NMVOC emissions from human activities in these areas were indeed associated with demand of export rather than local or domestic demand (Fig. 1a). Considering only the trade-relevant emissions, export emissions in Guangdong, Fujian, Shanghai, Shandong and Zhejiang indeed amounted to 44%, 41%, 36%, 33% and 30% of the emissions driven by interregional trade within China, respectively. In addition, it should be noted that NMVOCs as a group include hundreds of species. Among the export-related emissions in the above areas, around 20–35% were in the forms of BTEX, which have well-documented influences on the central nervous system and immune functions. Benzene and

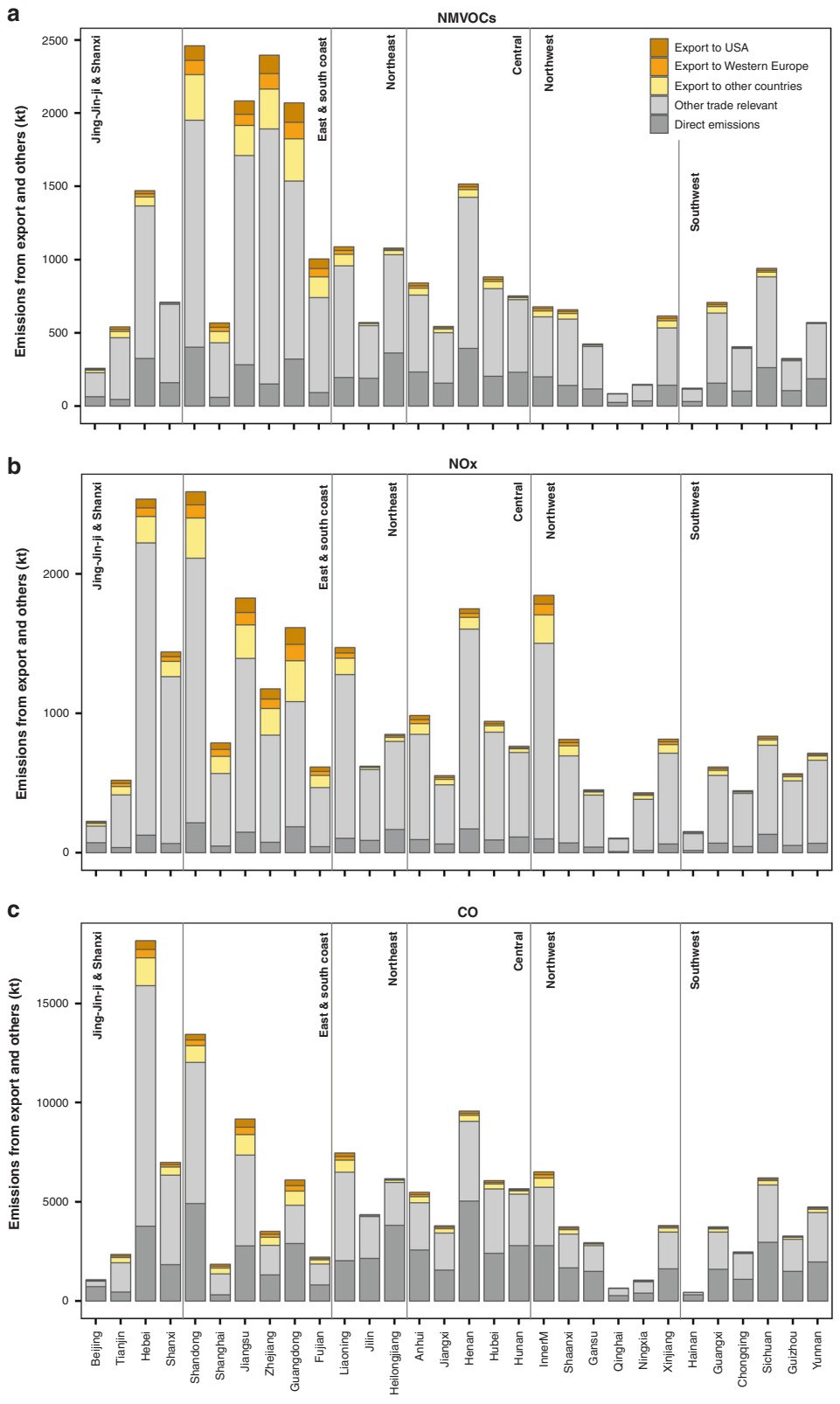

**Fig. 1 China's provincial emissions driven by export and domestic demands in 2013. a** NMVOCs. Export-driven emissions stood out in the east and south coast, e.g., Shandong, Jiangsu, Zhejiang and Guangdong. Demands from the United States (USA) and western Europe explained nearly half the export-relevant emissions. **b** NOx. Export contributed to 15% national sum of NOx emissions. Export-embodied emissions were notable in the east and south coast, as well as other inland provinces such as Inner Mongolia and Hebei. **c** CO. About 10% CO emissions in China were driven by export. Impacts of export were highlighted in Hebei, Shandong and Jiangsu. Source data are provided as a Source Data file.

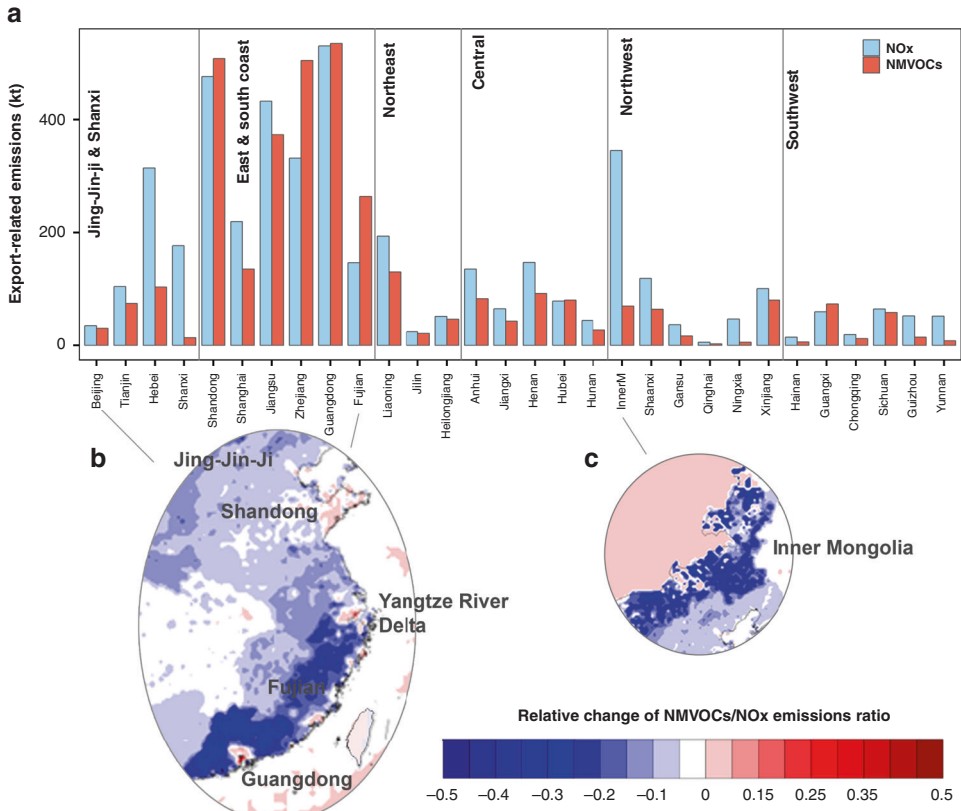

**Fig. 2 Relative emissions of NMVOCs and NOx alerted by export demand. a** Absolute value of NOx (in blue) and NMVOCs (in red) related to export demand by provinces. **b** Relative change in the ratio between anthropogenic NMVOCs to NOx emissions in the east and south coast in July. Blue indicates that export demand decreases the NMVOC/NOx ratio and red for increase. Export generally increased the NMVOC/NOx ratio in urban areas or industrial sites, but decreased the ratio in rural areas. **c** Export significantly decreased the NMVOC/NOx ratio in the Inner Mongolia since export demand in this province was largely associated with activities from power supply sectors to support the production of export goods with intensive NOx emissions. The relative changes in other provinces are generally less than 10% and not shown here. Source data are provided as a Source Data file.

ethylbenzene are even classified as Group 1 and Group 2B carcinogens by the International Agency for Research on Cancer (IARC)[20,21], respectively. The impact of export on the environmental concentration of BTEX will be explored in the next section. Regarding NOx and CO, the contributions of export were generally higher in the coastal areas of export industry hubs as well as some inland provinces such as Inner Mongolia and Hebei (Fig. 1b, c). This reflects the emission characteristics of NMVOCs, NOx and CO and the industry layout of China. In addition to common sources such as transportation, NOx and CO emissions are generally from fossil fuel combustion from the energy-intensive and heavy industrial sectors, while NMVOC emissions are emitted from miscellaneous non-combustion processes of light industries. As light industries thrive in the east and south coast, but heavy industries in the northern and inland provinces, the NMVOC emissions from export in coastal provinces generally outweighed those of NOx emissions, and vice versa for northern and inland provinces (Fig. 2a).

Tropospheric O$_3$ distinguishes from other air pollution partly because of its non-linear relationship with NMVOCs and NOx emissions. The emission ratio of NMVOCs and NOx largely determines the O$_3$ formation chemistry and hence the effectiveness of air pollution mitigation strategy[4,17]. By altering the emissions of NMVOCs and NOx disproportionately across the country (Fig. 2a), the demand of export has shaped the O$_3$ chemistry in the ground level in a hidden way. For the vast majority of China, export decreased the NMVOCs to NOx emission ratios by around 5–10%. In the suburban and rural areas

along the coast and some energy-supplying inland provinces such as Inner Mongolia (Fig. 2b, c), export had nearly cut down the NMVOC/NOx ratio by 50%. The urban and industrial areas in the export industry hubs, such as Guangdong, Fujian, Yangtze River Delta and Shandong, were a few exceptions that saw an increase in NMVOC/NOx ratio (Fig. 2b). These areas are largely overlapped with China's O$_3$ hotspots[3] and are governed by a NMVOC-limited or transitional regime[4,11,22]. Activities to support export have indeed made the O$_3$ regime less NMVOC-sensitive. For the other vast areas with decreased NMVOC/NOx ratio, however, they are governed by a NOx-limited regime[22], and the O$_3$ sensitivity to NOx emissions would be even higher without the export demand. Considering the rise of South–South trade, there is an emerging trend of production activities relocating from China and India to other developing countries[9]. If such relocations are significant enough, China should prepare for the changes in not only the scale of O$_3$ precursor emissions but also the O$_3$ chemistry. The current NMVOC-limited or transitional regime in the urban and industrial areas along the coast would be more limited by NMVOC emissions, while the NOx-limited regime in the majority of China would be enhanced.

**Changes in primary and secondary pollution due to export demands.** All the export-relevant emissions of NMVOCs, NOx and CO were excluded from the air-quality model (Case 1, detailed settings can be found in Supplementary Note 1) and compared with base case to study the impact of export demands

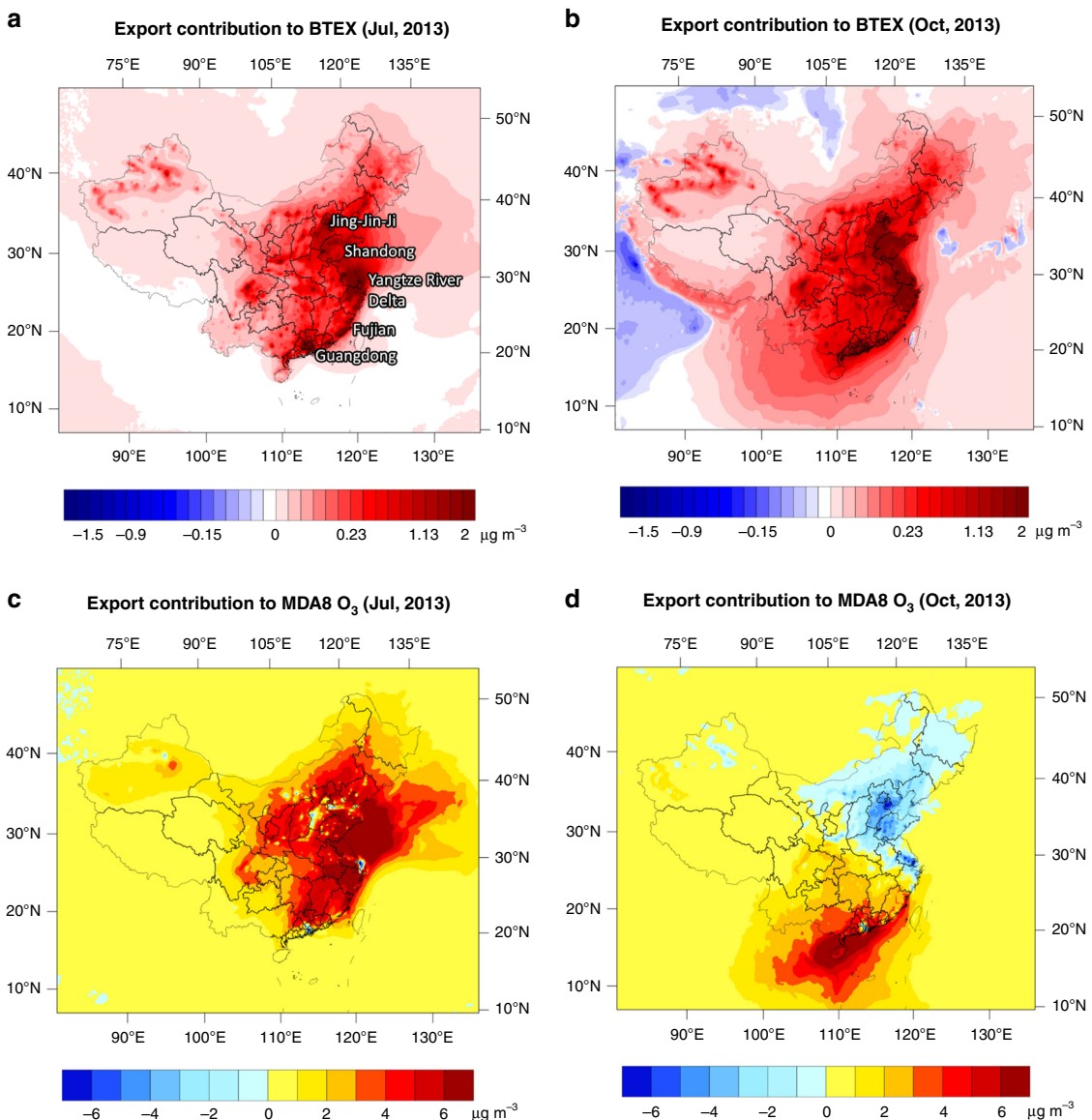

**Fig. 3 Export contributions to ambient BTEX and O$_3$ concentrations in China. a, b** Export increased the BTEX concentrations all year round in China, especially in the North China Plain and coastal areas (>0.5 μg m$^{-3}$). **c** Export elevated the peak O$_3$ level in July by 2–6 μg m$^{-3}$, varying from provinces. **d** Effects from export were mixed in October. While export emissions still contributed positively to the O$_3$ formation in south China, it inhibited peak O$_3$ by 1–3 μg m$^{-3}$ in the vast areas north to the Yangtze River Delta, especially the Jing-Jin-Ji area (3–5 μg m$^{-3}$). Source data are provided as a Source Data file.

on the ambient concentrations of BTEX and O$_3$. As primary pollutants, changes in BTEX concentrations due to export were consistent across different seasons (Fig. 3a, b). The coastal areas in China had suffered from an increase in BTEX concentrations of 20–30% annually. It resulted in an estimated 15,707 (3488–28,671, 95% CI) premature deaths per year, considering only the short-term environmental exposure.

The effects of export on China's O$_3$ concentration varied from seasons and latitudes. In July, which is the typical O$_3$ season in the north China and the Yangtze River Delta, the impact of export was generally consistent across the country. It had elevated the maximum daily 8-h average (MDA8) O$_3$ in the vast majority of China by 2–3 μg m$^{-3}$ (Fig. 4c). The effect of export stood out in the coastal areas of Shandong, Jiangsu, Zhejiang and Fujian with an increase of more than 6 μg m$^{-3}$, or 6–20% peak O$_3$ level by anthropogenic causes. In October, the impact of export-related emissions varied. Similar increase in O$_3$ was observed in southern China (red and orange areas in Fig. 4d). In the vast areas north to

the Yangtze River Delta, instead of increase, export-related emissions had inhibited the O$_3$ level (blue areas in Fig. 4d). Inhibition from export emissions was most notable around the Jing-Jin-Ji area, Shandong and Zhejiang (3–5 μg m$^{-3}$). Similar inhibition effects are observed for the other non-O$_3$ seasons, such as January and April (see Supplementary Fig. 2).

Such inhibition effects are mainly attributed to the temporal and spatial variations of O$_3$ formation regimes in China. In July, the vast majority of China is governed by a NOx-limited chemistry with a few exceptions in very limited areas of Jing-Jin-Ji, Yangtze River Delta and Guangdong. Under the NOx-limited regime, the increase in NOx and NMVOC emissions introduced by export demand could lead to a growth of maximum O$_3$ concentration. As the temperature dropped from July to October, biogenic NMVOC emissions declined dramatically and drove the O$_3$ regime towards NMVOC-sensitive. This is especially true for northern provinces where temperature dropped more significantly than that in the south. On top of this, the demand of

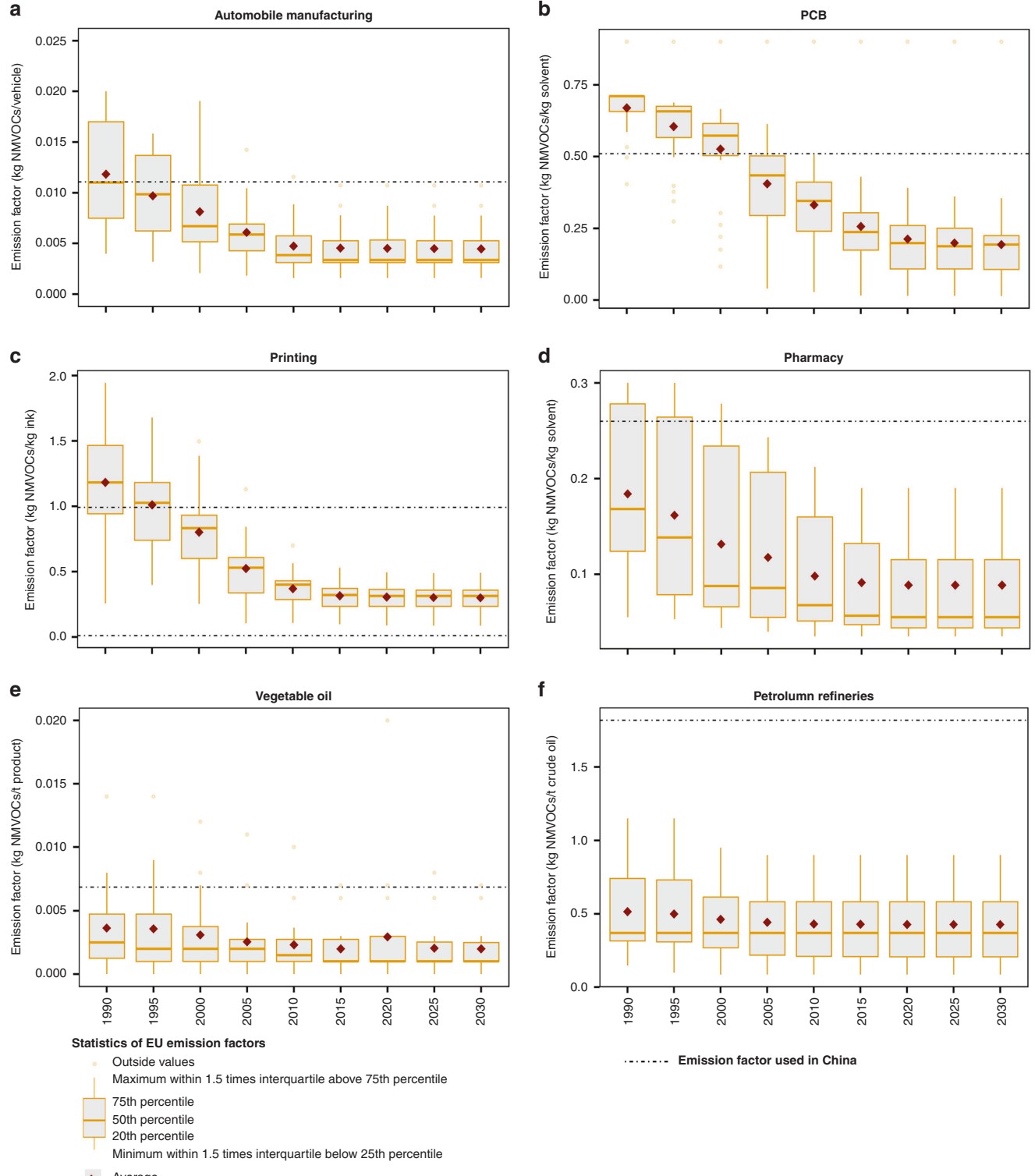

**Fig. 4 Gaps of NMVOC emission intensities between China and EU.** Boxplots represent the distribution of European levels in every 5 years from 1990 to 2030. The dotted line denotes the level of China in 2013. Intensities of China fall within the range of EU for most sectors, e.g., **a** automobile manufacturing, **b** PCB, **c** printing, **d** pharmacy and **e** vegetable oil. For **f**, petroleum refineries, intensities in China were symmetrically higher than in Europe. For evaluation of emission-reduction potentials, relative change in Europe was adopted instead of absolute value. Source data are provided as a Source Data file.

exports had pumped more NOx than NMVOC emissions in the atmosphere in most areas (as discussed in the last section). With a NMVOC-limited regime, an increase in NOx emissions could result in the decline of OH radical pool to react with NMVOCs and consequently inhibited the formation of $O_3$.

To sum up, emissions from export contributed positively to the peak $O_3$ level in the $O_3$ seasons, but help alleviate the low-level $O_3$ in other months. In this sense, export contributed to the exceedance rates of MDA8 $O_3$ in China, but its health burden would be less. Given the limited evidence that a threshold exists

for the association between exposure to O₃ and the risk of death[23–26], low-level $O_3$ could pose negative health impacts. By alleviating the low-level $O_3$ in non-$O_3$ seasons, part of the health burden introduced by export was offset and resulted in annualised premature deaths of 1182 (351–1,992, 95% CI).

**Closing the gap in emission intensity**. The adverse impact from export activities can be potentially eased by either decreasing the quantity of export goods or cutting down the emissions emitted per unit of goods ('emission intensity'). The ongoing US–China trade war and the emerging South–South trade overshadow the future of China's export industries. While it is difficult to predict precisely how the export industries will develop, we analysed the emission-reduction potentials of China's export industries and their impacts on the whole production capacity under static conditions (the 2013 productivities). Given the fact that NOx and CO emissions have been reduced aggressively under the clean-air actions in China[14], the focus was on NMVOC emissions from export-relevant industries. Specifically, they are the 20 industrial sectors with high volumes of export goods (Table 1). Considering the availabilities of sectoral emission

factors and the corresponding technical coefficients, NMVOC emission levels per unit of goods produced in China were compared with those in the EU as estimated in the GAINS model[27], which were assumed to represent the cleaner production practices with proven and affordable technologies[28]. For most industrial sectors, the emission intensities in China around 2013 were comparable to the upper bound of the EU around 2000, as shown in Fig. 4a–e. Following the experience in EU, NMVOC levels can be substantially cut down. For example, it has been shown that implementation of improved management practices in degreasing sector could lead to 41% lower emissions, while more advanced techniques, such as cold cleaner and the combination of sealed degreasers and activated carbon adsorption, can reduce emissions by well over 90%. For a few sectors such as petroleum refineries and rubber-tyre production, the emission levels in China are systematically higher than those in the EU. This might be attributed to the different compositions of products or poorer management along the production line that leads to higher NMVOC emissions.

By benchmarking the emission levels in China with those attainable in EU by 2030, reduction potential for China's export

**Table 1 Potentials of emission reductions and abatement cost.**

| Sources | NMVOC emission factors | | | NMVOC- reduction potentials (ton)[a] | Annualised abatement cost (million $) | Industrial output in 2013 (million $) |
|---|---|---|---|---|---|---|
| | China in 2013 | Possible low level | Unit | | | |
| Petroleum refinery | 1.82[b] | 1.08[c] | kg t⁻¹ product | 353,972 | 0 | NAᵖ |
| Extraction of edible oil | 6.88[d] | 2.29[c] | kg t⁻¹ product | 256,444 | 0 | NAᵖ |
| Tyre | 0.6[e] | 0.44[c] | kg piece⁻¹ | 176,464 | 0 | NAᵖ |
| Wood-furniture making | 0.92[f] | 0.49[c] | kg piece⁻¹ | 252,913 | 0 | NAᵖ |
| Extraction of oil | 1.42[b] | 0.93[c] | kg t⁻¹ product | 105,982 | 0 | NAᵖ |
| Paint manufacturing | 15[b] | 11[c] | kg t⁻¹ product | 38,170 | 35 | 53,226 (0.24%) |
| Ink manufacturing | 50[b] | 36[c] | kg t⁻¹ product | 9459 | 9 | |
| Dye manufacturing | 81[b] | 58[c] | kg t⁻¹ product | 20,575 | 19 | |
| Carbon-black manufacturing | 52[b] | 37[c] | kg t⁻¹ product | 69,036 | 64 | |
| Glue manufacturing | 11[g] | 8[c] | kg t⁻¹ product | 15,742 | 15 | NAᵖ |
| Printing | 993[f] | 301[c] | kg t⁻¹ ink | 396,216 | 501 | 167,718 (0.30%) |
| Shoe making | 0.028[f] | 0.020[c] | kg pair⁻¹ | 37,087 | 56 | 106,140 (0.05%) |
| Printed circuit board | 0.22[h] | 0.09[c] | kg m⁻² product | 29,019 | 55 | 22,548 (0.24%) |
| Metal coating (small devices) | 0.20[b] | 0.08[c] | kg piece⁻¹ | 67,305 | 127 | NAᵖ |
| Metal coating (large devices) | 0.40[b] | 0.15[c] | kg piece⁻¹ | 216 | 0.4ᵒ | NAᵖ |
| Pharmacy | 260[i] | 125[c] | kg t⁻¹ product | 354,546 | 977 | 359,629 (0.27%) |
| *Automobile manufacturing* | | | | | | |
| Bikes | 0.3[b] | 0.12[c] | kg VEH⁻¹ | 4290 | 26 | 853,225 (0.13%) |
| Small vehicles | 2.43[b] | 0.972[c] | kg VEH⁻¹ | 19,208 | 115 | |
| Other vehicles | 21.2[b] | 8.48[c] | kg VEH⁻¹ | 152,106 | 911 | |
| Motorbikes | 1.8[b] | 0.72[c] | kg VEH⁻¹ | 13,470 | 81 | |
| Coking | 2.1[j] | 0.427[k] | kg t⁻¹ coal charged | 1,128,867 | NA | NAᵖ |
| Polymeric coating | 0.182[b] | 0.009[l] | kg m⁻² surface | 818,404 | NA | NAᵖ |
| *Polymers and resins* | | | | | | |
| Polyethylene | 7.85[m] | 2.00[k] | kg t⁻¹ product | 68,679 | NA | NAᵖ |
| Polypropylene | 3.00[b] | 0.35[n] | kg t⁻¹ product | 33,019 | NA | NAᵖ |
| Polyvinyl chloride | 0.7448[b] | 0.1[k] | kg t⁻¹ product | 9865 | NA | NAᵖ |
| Polystyrene | 2.92[b] | 0.15[k] | kg t⁻¹ product | 5817 | NA | NAᵖ |

aReduction potentials estimated based on the activity level in 2013.
bEmission factor from MEP, P.R. China[49].
cValue is estimated based on the EU-average emission-factor trajectory.
dWeighted average of the emission factors of corn oil, cottonseed oil, peanut oil and soybean oil from MEP, P.R. China[49].
eAverage factor of MEP, P.R. China (2014)[49] and previous studies[43,47,50,51].
fFrom a field survey in the Pearl River Delta[52]. The factor of printing is the average of offset printing, rotogravure printing and letterpress printing.
gLocal factor unavailable. Factor from EMEP/EEA[53].
hFrom a field survey in the Pearl River Delta[54].
iEmission factor from Zheng et al.[14].
jLocal factor unavailable. Factor from US EPA (2008)[55] was adopted[50,56,57], which was based on the higher bound of emission level in an earlier study by Economic Commission for Europe (ECE)[58].
kBased on the lower bound of emission level by ECE[58].
lBy carbon adsorption units using activated carbon, 95% of NMVOCs from this process can be removed[55].
mAverage of high- and low-density polyethylene emission factors from MEP, P.R. China[49].
nFactor from US EPA (2008)[55].
oThe value might be underestimated since only the activity-level data of the cutting machine was available from the national statistics.
pThe industrial outputs for these sectors were not available since they were integrated with other sectors in China's official statistical systems.

industries was estimated. For those sectors with emission intensities within the EU range, the average level across the EU countries (instead of the median or country with the lowest value) was used as a reference for the possible low level that can be achieved. For the few sectors with systematically higher emission levels, a relative change was adopted instead of an absolute value. It is estimated that 57% of NMVOC emissions from export industries (excluding transport and other supporting activities) could be reduced (1165 kt). When these 1165 kt of NMVOCs were excluded from export industries and NOx and CO remained constant (Case 2, detailed settings can be found in Supplementary Note 1), a nationwide decrease in BTEX and MDA8 $O_3$ is observed, especially along the coastal areas (see Supplementary Fig. 3a–d). The export footprint in terms of health burden can be reduced by 37% by saving 6,520 (1396–11,954, 95% CI) premature deaths associated with the exposure of BTEX and MDA8 $O_3$ on an annual basis.

In reality, the assumption that cleaner production practices in a sector are only applied to export goods is frail and hardly plausible. If a shoe-making factory decided to upgrade its technology and resource management, it is more plausibly done for the whole production line rather than only for the shoes for export. Expanding the efforts from export sectors to the whole production capacity, a reduction of 4437 kt of NMVOCs would be expected, i.e., 58 and 17% of industrial and total anthropogenic NMVOC emissions in China, respectively. Considering the challenges in controlling NMVOC emissions (+11% from 2010 to 2017) and the rising relative contribution from industry, such decreases entail significances in China's NMVOC control. As shown in Fig. 5a, b, the reduced NMVOC emissions would lead to more than 0.5 μg m$^{-3}$ decrease of BTEX concentration in most areas (Case 3, detailed settings can be found in Supplementary Note 1). For Jing-Jin-Ji, Shandong, Yangtze River Delta and Guangdong, the decrease was more than 1 μg m$^{-3}$, accounting for

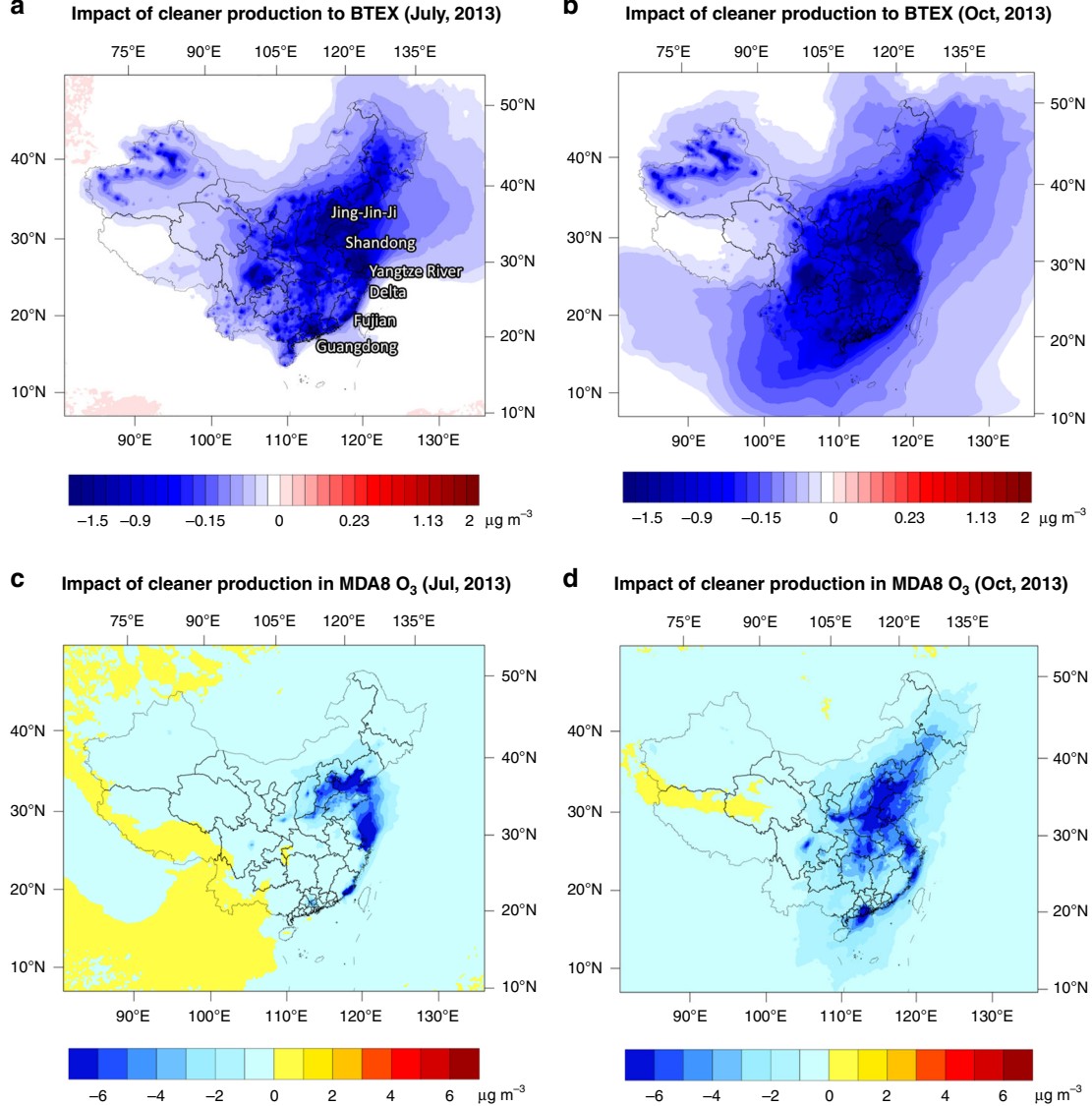

**Fig. 5 Efficacy of cleaner production manners in whole production capacities. a, b** BTEX decreased by more than 0.5 μg m$^{-3}$ in most areas when emission intensities of NMVOCs from key industrial sectors were lowered to European references under the 2013 emission rates. Coastal areas and the North China Plain experienced a decrease of 20–35% (>1 μg m$^{-3}$). **c, d** Cleaner production manners also resulted in a nationwide decrease in MDA8 $O_3$ under the 2013 emission rates, especially for the east and south coast and the North China plain (2–8 μg m$^{-3}$ in $O_3$ seasons). Source data are provided as a Source Data file.

20–35% of the BTEX concentration in 2013. Similarly, a nationwide decrease of peak $O_3$ in both $O_3$ and non-$O_3$ seasons was observed, especially for the east and south coast and north China plain (2–8 µg m$^{-3}$ in $O_3$ seasons, Fig. 5b, c). To put such reductions in context, one should note the challenge to reduce peak $O_3$ level. Considering the Yangtze River Delta as an example, an 8 µg m$^{-3}$ decrease in MDA8 $O_3$ would require a reduction of either 35% of NOx emissions or 32% of NMVOC emissions[11]. For Pearl River Delta (PRD), it would be a reduction of either 45% of NOx or NMVOC emissions[11]. Nationally, such reductions could save 28,309 people (6236–51,495, 95% CI) dying prematurely from diseases associated with BTEX and MDA8 $O_3$ per year.

The costs for introducing such low-emission practices were estimated at 0.05–0.3% of the annual industrial output, varying across sectors (Table 1). For pigment-manufacturing and shoe-making industries, emissions can be cut down by around 30% with annualised costs of 0.24% and 0.05%, respectively. Regarding printing, PCB, pharmacy and automobile manufacturing, sectoral emission reduction of 50–70% can be achieved with annualised cost from 0.13 to 0.3%. Negative unit costs were estimated for few sectors such as tyre manufacturing, wood-furniture making and extraction of edible oil. It is because the value of saved or recovered solvent (e.g., hexane in vegetable oil-producing process) offsets the investment and additional operating costs of control technologies. The recovery of these NMVOCs does not only reduce the emissions, but also increases the output and revenue. Since prices of solvents, pollution-discharge fees, labour costs and other input material costs are generally lower in China, negative costs estimated here might be overstated. Nevertheless, the 'true' costs for these sectors should not be excessive and decrease over time. Therefore, we assume that the costs are relatively low and set them as zero in Table 1.

The estimated cost is comparable to a study in the PRD, South China[29]. Costs for adsorption by activated carbon and a switch from low-solvent to solvent-free paints were estimated as $501 and $13317 per ton of abated NMVOCs in that study, respectively. The cost for solvent substitution is much higher. Estimated costs in this paper fall within the above range, varying from $923 to $5992 per ton of NMVOCs; the upper bound is lower than that of the previous study since a mix of technological means is adopted in each sector. For instance, a combination of process modification, solvent substitution, adsorption and incineration techniques are adopted in the automobile- manufacturing sector. As a result, the average cost would be lower than a sole measure of solvent substitution.

**Efforts from homeland and abroad to reduce $O_3$ and its precursors**. By filling the gap of consumption-based $O_3$ study, this study reveals the responsibilities of global consumers and opportunities for future efforts from homeland and abroad to address the tropospheric $O_3$ problem. China exports goods to 140 countries or regions in the world[30]. Among them, export to USA alone accounted for 22%, 20% and 20% of the NMVOCs, NOx and CO emissions embodied in China's export goods, respectively. Western European countries (see the country list in Supplementary Table 1) together explained around 20% of the export-driven NMVOCs, NOx and CO emissions in China. Another share of 20% was associated with the demands in developed regions in Asia and Pacific. Consumption in the above three regions each entails approximately 3700 premature deaths per year in China due to the elevation of BTEX and MDA8 $O_3$.

The export footprint can be reduced by more responsible consumption and production. By reducing the disposal of products within their service lives and increasing recycling, a large part of the consumption of electrical equipment, metal devices, furniture, shoes and leather products and others can be avoided. This might be especially applicable for the developed countries that together accounted for over 60% of the export goods from China and excess consumption exists. Many consumers are not fully aware of the environmental footprints of the products they consumed or only focus on the $CO_2$ footprints. To enable a fundamental shift in demand side, a transparent system in the embodied environmental impacts of products should be established. From the perspective of production, there exist great potentials in reducing the export footprints by accelerating technology transfer. Emissions of $O_3$ precursors in production per unit of export goods in China are still consistently higher than those in European countries with stricter emission standards. With the proven and affordable technologies, export footprint can be reduced by ~40%. Analysis of mitigation costs indicates that NMVOCs from export sectors can be reduced by around 60% at the expense of less than 0.3% of the annual industrial output. The price competiveness of export goods would not be seriously undermined. Despite the current setback of technology transfer under the clean development mechanism (CDM), global traders should be made aware of the significant benefits of cleaner production technology and management in terms of environmental impacts, and explore the potential for future collaboration.

Even for the world's top exporting countries like China, production for the domestic market still needs to be addressed to achieve substantial reduction of NMVOC emissions and $O_3$. The direct and indirect consumption of urban and rural households in China contributes about 40% of NMVOC emissions. With increasing household income and consumption, that contribution is expected to grow further. Policies addressing household products and consumer behaviour should be formulated. Long-term attainment of $O_3$ across the country would also call for further $NO_x$ reduction of >50%[30]. As demand from abroad accounted for about 15% of China's $NO_x$ emissions in 2013, strategies targeting domestic demand driving $NO_x$ emissions and end-of-pipe treatment would be the key to halve NOx emission and consequently bring ambient $O_3$ to a safe level nationwide.

This study reveals the complex interplay between exogenous demands and the formation of tropospheric $O_3$ in China. Due to the non-linear relationship between NMVOCs, NOx and $O_3$ and the other contributing factors such as sunlight and temperature, export emissions have mixed impacts on the tropospheric $O_3$. They have inhibited the $O_3$ formation in non-$O_3$ seasons, especially in the areas with higher latitude and notable seasonal changes of sunlight and temperature (e.g., the North China Plain). Such a relationship should be investigated in other countries to reveal how the global supply chain has shaped the tropospheric $O_3$ globally.

Another concern is the emerging trend of the relocation of global supply chain and whether it will exacerbate the existing $O_3$ pollution in the low- and mid-latitudes. A few factors are driving the relocation of the global supply chain. One is the rise of South–South trade, and it is reported that some of China's export capacities had moved out to other emerging economies[9]. The other is the ongoing US–China trade war. A large proportion of industrial products characterised by high NMVOC emission intensity are subject to recently increased tariffs, such as paints, dyes, glues, adhesives, wood furniture, man-made textiles, machinery, electronics, vehicles and parts, ships and boats[31]. These two factors might be accelerated under the COVID-19 crisis due to the disruptions to supply chains, and more companies are working through alternative sourcing strategies. While a small amount of the capacities might move back to developed economies, south Asian and African countries will be

the major receptors of the relocated export capacities. For lower-latitude areas in these regions, the inhibition effects observed in latitudes similar to China's North China Plain might not be applicable. A full-year increase in $O_3$ might be observed, and the health burden would be much higher than what we reported here for China. Tropospheric $O_3$ and its precursors in these exporting countries should be monitored closely to avoid severe disruptions in human health and crop yields[32].

## Methods

This study was conducted applying a validated air-quality modelling platform with emission inputs from environmentally extended input–output (EEIO) analysis and emission-reduction scenarios. First, consumption-based emission inventories for NMVOCs, NOx and CO were developed from EEIO analysis. Emissions relevant to final demands including export were revealed. Second, export-relevant emissions were excluded from the air-quality model (Case 1) and compared with the base case to study the contribution of export demand on China's $O_3$ formation. Another two cases were then constructed to study the effectiveness of NMVOC reductions from merely the export industrial capacities (Case 2) and the whole capacities (Case 3). Finally, $O_3$ health-exposure relationship was applied to study the health cost or benefit from the above cases.

**Environmentally extended input–output analysis**. China's MRIO table for 30 provinces and 30 sectors was linked to the GTAP database to study the impact of export and the originating countries with the established methods for EEIO analysis[8,33]. The total outputs of sectors in a given economy ($X$) can be understood as the sum of the intermediate input to other sectors ($Z$) and the finished goods for final consumers ($Y$). For the global economy with $M$ regions and $N$ industries in each region, $x_i^r$ represents the total output of industry $i$ in country $r$ and can be expressed as

$$x_i^r = \sum_{s=1}^{M} \sum_{j=1}^{N} z_{ij}^{rs} + \sum_{s=1}^{M} y_i^{rs}, \tag{1}$$

where $z_{ij}^{rs}(r, s = 1, 2, …, M; i,j = 1, 2, … N)$ represents the intermediate product sold from industry $i$ in country $r$ to industry $j$ in country $s$, $y_i^{rs}$ represents the finished goods sold from industry $i$ in country $r$ to the final consumers in country $s$.

A technical coefficient $a_{ij}^{rs} = z_{ij}^{rs}/x_j^s$ is defined as the input from sector $i$ in region $r$ needed to produce one unit of output from sector $j$ in region $s$. Equation (1) can therefore be formulated as follows:

$$X = (I - A)^{-1}F, \tag{2}$$

where $X$, $A$, $F$ and $I$ are the matrices of $x_i^r$, $a_{ij}^{rs}$, $y_i^{rs}$ and an identity matrix, respectively.

To calculate the $O_3$ precursor emissions embodied in goods and services, emission intensity (i.e., $O_3$ precursor emissions per unit of economic output) is introduced. The NMVOCs, NOx and CO emissions embodied in goods and services can be calculated as

$$C = h(I - A)^{-1}F, \tag{3}$$

where $C$ is the matrix showing the emissions embodied in goods and services used for different final demands, and $h$ is a vector of emission intensity by sector and region.

**Air-quality modelling platform**. The study domain for this work is mainland China, with a spatial resolution of $27 \times 27$ km. The air-quality modelling platform coupled the Weather Research and Forecast (WRF) model[34], SparseMatrix Operator Kernel Emissions (SMOKE) model[35] and CMAQ model[36]. The Weather Research and Forecast (WRF) model v3.9 was used to provide meteorological data. The platform reproduced the $O_3$ pollution in January, April, July and October of 2013, representing the peak $O_3$ months (July and October) in the northern and southern China and supporting the health-risk analysis, respectively. Model performances were evaluated by ambient $O_3$ measurements (see Supplementary Note 2). The correlation coefficient ($R$) between the modelling result and observations was between 0.50 and 0.78 for 1-h average or maximum daily 8-h average, similar to those of previous studies in China[37–39]. Detailed model configurations and validation of CMAQ and WRF are shown in Supplementary Tables 2 and 3 and Fig. 4. Bulk emission inventories from the EEIO analysis were processed by the emission-processing module with localised temporal and spatial surrogates to have the model-ready emission inputs for simulation and analysis.

**Health-impact estimation**. Epidemiological studies on the detrimental effects of BTEX and ambient $O_3$ on population health are emerging in China. The concentration-response functions from studies within China are prioritised in this study. The number of premature deaths due to a change in ambient BTEX and $O_3$

concentration was estimated as follows:

$$\Delta M = \sum_{i=1}^{15321} POP_i \times M_i \times CRF \times \Delta X_i, \tag{4}$$

where $i$ is the number of 15321 grids ($27 \times 27$ km) in accordance with the modelling platform, $POP_i$ is the number of people in Grid $i$, $M_i$ is the baseline cause-specific mortality in Grid $i$, $CRF$ is the concentration-response function from epidemiological studies, $\Delta X_i$ is the change of BTEX or $O_3$ concentration in Grid $i$. Baseline cause-specific mortality for the year of 2013 was obtained from the burden of disease study in China[25] and the statistical yearbooks[40]. Since mortality data are only available at the provincial level, grids within the same province adopted the provincial value. CRF is pollutant- and disease-specific. For BTEX, the short-term increments of environmental benzene (1 μg m$^{-3}$) and TEX (1 μg m$^{-3}$) are associated with 4.1% (0.7–7.7%, 95% CI) and 0.44% (0.13–0.77%, 95% CI) increases in circulatory mortality[15]. For $O_3$, the pooled CRF associated with a 10 μg/m$^3$ increase in $O_3$ concentrations from eight epidemiological studies in China is 0.60% (0.22–0.97%, 95% CI) and 0.51% (0.03–0.98%, 95% CI) for cardiovascular and respiratory diseases, respectively[25].

**Data sources**. The MRIO table was adopted from previous studies[7,8,33]. It was developed based on the 2012 Chinese national and provincial single-region input–output tables. The table has been demonstrated and used to study the driving demands and trade-related contributions to greenhouse gases[8,33,41], and some of the air pollutants in China[7]. GTAP version 9 was used, which described bilateral trade between 140 regions for 57 sectors. The production-based emission inventories of NMVOCs, NOx and CO for China were developed based on the established methodologies, and the best available local emission factors[14,42–47], for the base year of 2013. For China, anthropogenic NMVOC emissions from stationary combustion, on-road and non-road mobile sources, industrial processes, industrial and household solvent use, biomass burning and others (such as gas stations and dry cleaning) were estimated. Activity-level data were collected from national and provincial statistical yearbooks. As for other countries in the world, NMVOC emissions were taken from the Emissions Database for Global Atmospheric Research (EDGAR) v4.3.2 database[19]. Biogenic VOC emissions were estimated by Model of Emissions of Gases and Aerosols from Nature (MEGAN)[48]. It should be noted that there was a 1-year gap in the MRIO table (in 2012) and production-based emission inventories (in 2013). The MRIO tables heavily rely on the statistics of trade flow, which is only available for 2007, 2010 and 2012 in provincial levels. But nationwide measurements on ambient $O_3$ in China were not available until 2013. To enable the validation of the modelling platform, the reference year of emission inventories was set for 2013 instead of 2012. Nevertheless, the 1-year gap should not undermine the reliability of this study since we assume no dramatic change of trade characteristics from 2012 to 2013. Cause-specific mortality rate was obtained from China Health and Family Planning Statistical Yearbook[40].

## Data availability

The China Multi-Regional Input–Output Table 2012 can be downloaded from the China Emission Accounts and Datasets (CEADS) website (http://www.ceads.net/)[8,9]. Source data are provided with this paper.

## Code availability

Code used in this study is provided in a repository in Github (https://github.com/JmmOo/Consumption-based-study-on-O3-and-its-precursors-in-China).

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

## Acknowledgements

This work is supported by National Key Research and Development Program of China (No. 2018YFC0213904), and National Natural Science Foundation of China (No. 41921005, 91846301 and 71904007), the UK Natural Environment Research Council (NE/N00714X/1 and NE/P019900/1), the Economic and Social Research Council

(ES/L016028/1), and British Academy (NAFR2180103, NAFR2180104). The authors thank the support of Natural Environment Research Council, part of UK Research and Innovation, for Jiamin Ou's participation in the Young Scientists Summer Programme in IIASA (https://iiasa.ac.at/web/home/education/yssp/Young_Scientists_Summer_Program.html) and the support of the 2018 Mihalevich award by IIASA.

## Author contributions

J.O., Z.W., Z.K., J.Z. and D.G. designed the study. J.O. and Z.W. prepared emission profiles, performed analysis and prepared the paper. J.G. built the simulation platform. S.Z., C.L. and P.L. supported the emission inventory development. J.M., Z.M., H.Z. and Y.S. supported the EEIO analysis. Z.K., J.Z. and D.G. jointly coordinated and supervised the project.

## Competing interests

The authors declare no competing interests.
