## [Peer Review File · Nature Communications]

Reviewers' comments:

Reviewer #1 (Remarks to the Author):

This paper has found the relation between ozone concentration in China and Chinese export demands by connecting an input-output analysis (IOA) and air quality model, meteorological model, and exposure-risk model. These approaches are excellent and would provide reliable facts. However, there are two critical points where I am unable to agree. One is the paper focuses on export, and another one is a shallow discussion about the responsibilities of import countries. Specific comments are listed below.

L24: What CI percentage does the rage show, 95%?

L59-61: Main economic drivers of air pollutants emissions are not export, but domestic final demand. These sentences seem countermeasures on export has a priority and sound very misleading. Rational reason to focus on export should be provided.

L84-85: The contribution of each domestic final demand as households, government, fixed-capital to induced NMVOC should be shown first. Then, the authors should give a rational reason why export is a problem.

L93-94: Around 18-26% is a limited contribution. Why don't you focus on interregional trade in China?

L111-112: If multiple regions have reduced their emissions at the same time, how the ratio of NO_x/NMVOC will change? Can we estimate it from the figure2? Readers may be interested in which regions should have countermeasures preferentially, not as single but multiple regions.

L150-151: Why has the EU28 been selected, not but the US, the UK, Canada, Japan, or Australia? Is the EU the lowest region of ozone concentration?

L223-224: What is the responsibility of the US and other import countries? Consumption-based accounting is a concept to encourage import countries to address environmental problems together with producers. A little discussion on the importers' role may be disappointing readers.

L276: How have final demands for July and October been set to produce emissions in two seasons?

L310-320: Are emissions from gas stations and dry cleaning included?

Reviewer #2 (Remarks to the Author):

Review of "Role of export industries on ozone pollution in China" by Ou et al.

The paper examines the contributions of export industries and non-methane volatile organic compound (NMVOC) emissions to surface ozone pollution in China. The work is generally sound, and the writing is good, but I found the organization and explanation lacking in some key aspects.

The two topics of this paper are only very loosely related: the effects of emissions from export industries and costs and benefits of reduced NMVOC emissions. Combining them into one paper currently results in confusion. The abstract and introduction focus only on the effects of NMVOC emissions, which is the second topic. They do not mention the results of the first topic at all, which also perturbed NO_x and CO emissions as well as NMVOC emissions. As a result, it is very unclear to the reader what the processes are being tested in the simulations.

For the analysis of export industries, it's not clear if the emissions were changed separately for every industry in every Chinese province or if a single scale factor was applied to all sector emissions from each province.

The mortality analysis is basic. State of the art estimates of mortality from air pollution consider cause-specific mortality, rather than all-cause mortality, which varies tremendously based on the underlying reasons that people are dying. In addition, it isn't clear if the mortality analysis assumes uniform baseline mortality across all of China, or if the mortality rates used in the calculation vary spatially.

Lesser issues

The paper uses “~” when specifying a range of values e.g. lines 22, 23, 24, , 33, 34 and many more. This is unconventional.

The paper relies on Chinese government reports for background information on O3 and PM levels, specifically references 1, 2, 6. These are in Chinese and it is not clear to me if they are peer reviewed. Link in reference 2 does not work.

References 7,8, 27 are missing journal or book titles.

Reference 12 is about heterogeneous chemistry, and therefore not an appropriate citation for the role of NOx vs. NMVOC emissions ratios in controlling O3 chemistry.

Line 111: “Decrease was more notable in most O3 hotspots in China such as the Jing-Jin-Ji, Shanxi, Guangdong and Jiangsu (Figure 2). It suggests that demand of export have slightly increased the sensitivity of O3 formation to NMVOCs emissions (‘more NMVOCs-sensitive’).” This is not convincing. A NMVOC/NOx ratio around 1 would generally fall into the NOx-limited regime according to most classic photochemical models (e.g. Lin and Trainer 1988; Sillman et al., 1990). Satellite data also suggest that very little of China or Jing-Jin-Ji is in a VOC-limited regime, centered around urban areas (Jin and Holloway, 2015). A change from 0.94 to 0.91 in this ratio is not a meaningful change and would not expect to be associated with the region where O3 concentrations increase as NOx emissions decrease

Line 257 incomplete sentence.

Line 322 says “morality” instead of “mortality”

I do not have a basis for judging the reasonableness of the cost estimates in Table 1.

References

Jin X, Holloway T. 2015. Spatial and temporal variability of ozone sensitivity over China observed from the Ozone Monitoring Instrument. *J Geophys Res* 120: 7229–7246. doi: 10.1002/2015JD023250.

Lin X, Trainer M, Liu SC. 1988. On the nonlinearity of the tropospheric ozone production. *J Geophys Res* 93(D12): 15879–15888.

Sillman S, Logan JA, Wofsy SC. 1990. The sensitivity of ozone to nitrogen oxides and hydrocarbons in regional ozone episodes. *J Geophys Res* 95(D2): 1837–1851. doi: 10.1029/JD095iD02p01837

Response to Reviewers' Comments

Reviewer #1 (Remarks to the Author):

This paper has found the relation between ozone concentration in China and Chinese export demands by connecting an input-output analysis (IOA) and air quality model, meteorological model, and exposure-risk model. These approaches are excellent and would provide reliable facts. However, there are two critical points where I am unable to agree. One is the paper focuses on export, and another one is a shallow discussion about the responsibilities of import countries. Specific comments are listed below.

Response: We really appreciate the time and effort from the reviewer. The reviewer's comments, either positive or critical, have helped us a lot to improve the current manuscript. Regarding the first critical point, we admit that we did not make it clear why this paper focuses on the role of export. In addressing the specific comments raised by the reviewer, we justified why there is a necessity to study the role of export on China's tropospheric O₃ problem. Main text in the manuscript is revised accordingly to address the reviewer's concerns. For the second critical point, we restructured the last section of Results and Discussion to explore how importing and exporting countries can work together to reduce O₃ and its precursors.

L24: What CI percentage does the rage show, 95%?

Response: Thank you for pointing out this. Yes, it indicates the range of 95% confidence interval here. To make it clear, we have revised it as "(3,839-30,663, 95% CI)". Similar expressions in the manuscript have been checked and corrected throughout the manuscript (e.g., Line 34, 163, 193, 226 and 245).

L59-61: Main economic drivers of air pollutants emissions are not export, but domestic final demand. These sentences seem countermeasures on export has a priority and sound very misleading. Rational reason to focus on export should be provided.

Response: Many thanks for the comment! We totally understand the concern from the reviewer that the contribution from export seems to be not big enough. While it is true that main economic drivers of air pollutants emissions are not export but domestic final demands, it does not mean that we can overlook the emissions driven from exogenous demands (considering the fact that export-driven emissions are equivalent to half the emissions of 28 European countries). Indeed, there exists no pollutant in a country whose emissions from export could surpass those from

domestic demand. Previous studies revealed that production for export contributed to around 10% to 20% of the premature mortality attributable to PM_{2.5} in China, but still yield notable significances in terms of advancing the understanding of cross-boundary causes and responsibilities of air pollution [1]–[3].

We admit that the Introduction in the previous version did not make it clear why this study focuses on export. Therefore, we have rewritten the Introduction to provide a rationale of this study (please see in Line 40-103). An understanding of the role of export industries in China's O₃ pollution and its precursors has significances for China's persistent problems of O₃ and NMVOCs and advances the scientific efforts to characterize the cross-boundary causes and impacts of photochemical O₃. By focusing on export, this article wants to send out the message that the role of export industries in O₃ formation is far more complicated than what we have expected on (1) relocations of precursors emissions not only change the magnitude of emissions but also the relative emission ratios determining the O₃ chemistry (Line 137-156); (2) the mixed effects of export demands on China's high- and low-level O₃, and how they vary from seasons and latitudes (Line 165-193); (3) the health burdens introduced by not only O₃ but also its precursors (Line 157-164). We believe the focus of this paper helps readers rethink how the global supply chain has shaped the air pollution and what the future might look like (Line 303-324).

L84-85: The contribution of each domestic final demand as households, government, fixed-capital to induced NMVOC should be shown first. Then, the authors should give a rational reason why export is a problem.

Response: Many thanks for the suggestion! We have considered the possibilities to start the discussion from the contributions of households, government and capital formation. However, given that this is an interdisciplinary study that includes both environmental economics and air pollution chemistry, we decided to avoid using a lot of field-specific jargons. We have received feedback that the terms frequently used in input-output analysis, such as capital formation, are obscure for some readers. We would need additional space to just introduce these terms. To the readers, the emissions after deduction of export demands would naturally come down to domestic demands. More importantly, we do not want to distract the following discussion on export demand and would rather go straightforward into the contribution of export. The paragraph that the reviewer commented on has been revised as:

“The demand of export increases not only the production activities related to direct export products but also the activities from power supply sectors, transportation, and others to support the production of export goods. Nationally, export explains 13%, 15% and 10% of the NMVOCs, NO_x and CO emissions in 2013, respectively. As the world's largest exporter for a lot of VOC-relevant products, the contribution from export industries to national NMVOC emissions was not as high as expected.”

However, it does not mean that we can downplay the role of export. Though being dwarfed by China's total emission budget, export-driven NMVOCs emissions (3,500 kt in 2013) were equivalent to half the emissions of 28 European countries (EU-28) [13]. If such emissions were generated by a single country, it would be ranked as the 10th largest NMVOCs emitter in the world [14]." (Line 105-115)

We provided the first reason why we should not downplay the role of export here given the amount of emissions (Line 109-115). In the following paragraph (Line 115-137), we provided further reasons why export-driven emissions should be investigated. One is its much higher impact along the coastal areas. The other is the fact that BTEX emissions accounted for 20 to 35% of NMOVCs emissions driven by export. Indeed, they are only part of the reasons why we focus on export (refer to our response to the previous comment). Other important factors are incorporated in the later discussion as more results are revealing.

L93-94: Around 18-26% is a limited contribution. Why don't you focus on interregional trade in China?

Response: Thanks for the comment. We agree with the reviewer that interregional trade within China has higher contributions than export. And studies have been conducted to investigate the interregional inequality introduced by consumption [4], [5]. As we mentioned in the above response, however, we believe there are important data gap and messages we want to send out through this article. Export emissions naturally cannot match those driven by domestic demands. But for consumption-based study, it is a good starting point to understand the pollution for exporting countries such as China. Therefore we want to stay focused on export rather than extending the current discussion to other consumption drivers. However, we include brief discussion of impacts of policies of controlling export industries on the whole of China since we assume that these industries would produce more efficiently and cleaner for both domestic and international market.

Another reason why we did not focus on interregional trade is its policy implications. While policy implications for international export are quite clear in terms of the share of responsibilities between exporting and importing countries and international alliance, the same logic is not always applicable to interregional trade. While some interregional studies have been conducted on the regional inequality [4], [5], it is still a bit difficult to tell how they can benefit the existing domestic pollution alleviation (since provinces within China are already working closely to tackle pollution and the control measures are generally consistent across provinces under the central government). For sure, studies on interregional trade would also be enlightening and needed in future studies. But it would be too much for us to explore and discuss both elements in one paper.

Specifically, for the contribution that the reviewer mentioned here, we compared the export emissions to those associated with interregional trade within China. Nationally, export emissions are equivalent to 20% of those driven by interregional trade. For the export industry hubs, the figures rise to 30-44%: Guangdong (44%), Fujian (41%), Shanghai (36%), Shandong (33%) and Zhejiang (30%). If interregional trade should draw attention because of its size, then export-driven emissions which amount to 20-44% of its value should also be investigated. To address the reviewer's concern, discussion in the main text has been revised to include the above idea:

“Around 18-26% NMVOCs emissions from human activities in these areas were indeed associated with demand of export rather than local or domestic demand (Figure 1a). Considering only the trade-relevant emissions, export emissions in Guangdong, Fujian, Shanghai, Shandong, and Zhejiang indeed amounted to 44%, 41%, 36%, 33% and 30% of the emissions driven by interregional trade within China, respectively.” (Line 119-123)

We hope the review could agree that the current manuscript will still focus on export.

L111-112: If multiple regions have reduced their emissions at the same time, how the ratio of NO_x/NMVOC will change? Can we estimate it from the figure2? Readers may be interested in which regions should have countermeasures preferentially, not as single but multiple regions.

Response: Many thanks for the suggestion. Figure 2 is indeed generated when we compared the emissions in Case 1(that is excluding all the export emissions in all the provinces at the same time) to the base case (the 'real' emissions in 2013). Therefore, it reflects the changes of all regions instead of single region in terms of primary O₃ precursor emissions. Then the change of O₃ and environmental concentration of its precursors when the export-driven emissions in all regions were excluded were simulated in our modelling Case 1. Results are shown in Figure 3 in the current manuscript.

The reviewer may be suggesting the group of regions first, and then, the impact evaluation of their reduction. It should be noted that NMVOCs and NO_x are both primary pollutants with local impact. The reduction of multiple regions would mainly affect their emissions within their counties and Figure 2 reflects the changes of all regions. Moreover, different from production-based emissions, the consumption-based emissions in one province are highly intertwined with other provinces through the supply chain. It would be arbitrary or impractical to reduce only the emissions in a few provinces but keep the emissions same in others. For example, export-driven emissions in Inner Mongolia cannot be reduced without consistent reduction of the activities in Shandong or Yangtze River Delta. It is not plausible to assume that the production in Shandong and Yangtze River Delta continue yet they don't need

electricity from Inner Mongolia anymore (only if the energy supplying system changes dramatically within China). Though the multiple regions hypothesis might be interesting to explore for O₃ control, we are afraid that it would be too much for this manuscript to cover. It is due to the above reasons that we did not explore the possibilities of multiple region reduction.

L150-151: Why has the EU28 been selected, not but the US, the UK, Canada, Japan, or Australia? Is the EU the lowest region of ozone concentration?

Response: Thanks for the comment. The reason why we selected the EU28 (including UK before Brexit) in this study is due to the comprehensive dataset for European countries within the GAINS model, which provides great details in the time series of sectoral emission factors, the abatement technologies and their costs. EU does not have the lowest concentration of O₃ (which also depends on the meteorological factors for photochemical reaction) in the world, but they have a long history in NMVOCs control since 1990 under the Economic Commission for Europe [6] and a set of proven and affordable technologies and management practices. Therefore it is used in this study for comparison. Explanation is provided now in the manuscript:

“Considering the availabilities of sectoral emission factors and the corresponding technical coefficients, NMVOCs emission levels per unit of goods produced in China were compared with those in the EU28 as estimated in the GAINS model [7], which were assumed to represent the cleaner production practices with proven and affordable technologies [8].” (Line 203-206)

L223-224: What is the responsibility of the US and other import countries? Consumption-based accounting is a concept to encourage import countries to address environmental problems together with producers. A little discussion on the importers' role may be disappointing readers.

Response: Many thanks for the suggestion. We agree with the reviewer and a more in-depth discussion on the responsibilities of importing countries and how can they work with exporting countries. A new section “Efforts from homeland and abroad to reduce O₃ and its precursors” (Line 269-325) is added in the manuscript now.

The key points covered by this new section include (1) the responsibilities of leading importers such as USA, Western Europe, and developed regions in Asia and Pacific in terms of premature deaths associated with the elevation of BTEX and O₃. Each of the above importers is responsible for around 20% of the O₃ precursors emissions embodied in China's export goods, and as a result, leads to approximately 3700 premature deaths per year in China. (2) Importing countries can reduce their export footprint by more responsible consumption and fastening the international

collaboration in technology transfer. Reduced disposal of products within their service lives and increasing recycling, a large part of the consumption of electrical equipment, metal devices, furniture, shoes and leather products and others can be avoided. This might be especially applicable for the developed countries which together accounted for over 60% of the export goods from China and excess consumption exists. The emission gap between China and some other importing countries has been demonstrated in this study. Importers have an important role to play to fasten the technology transfer between importing and exporting countries to reduce the environmental impact of their consumption. Given the experiences and lessons learnt from the clean development mechanism (CDM), global traders should explore more efficient ways to fasten technology transfer. This would be beneficial to not only China but also other exporting countries.

L276: How have final demands for July and October been set to produce emissions in two seasons?

Response: Thank you for the question. This comes down to the development of bulk emission inventories and its temporal and spatial allocation. Usually emission inventories are expressed as annual emissions for a given country or region in different source categories. Such emission inventories are called as the bulk emission inventories. To satisfy the input for air quality models such as the CMAQ we used in this study, however, the bulk emissions need to be break down into much details in terms of temporal (e.g., monthly, hourly) and spatial (e.g., $0.5^{\circ} \times 0.5^{\circ}$) emissions. Such a process is called the temporal and spatial allocation of bulk emission inventories. It can be done by collecting the temporal and spatial surrogates and the help of emission processing systems such as Sparse Matrix Operator Kernel Emissions (SMOKE). Such a methodology is well-established and can be found in the peer-reviewed papers by our co-authors [9]–[12].

In this study, the annual bulk emission inventory for the base year of 2013 is first developed. Given that the input-output table is also in an annual basis, the bulk emission inventory is used as an input for the environmentally-extended input-output (EEIO) analysis to produce the consumption-based emission inventory. The consumption-based emission inventory reveals that how many emissions in each source category are associated with the demand of export. By excluding the emissions driven by export, a new bulk emission inventory is generated for Case 1, which is the case for us to study the impact of export. The new bulk emission inventory is processed with the temporal and spatial surrogates and emission processing systems to have the monthly, daily and even hourly emission inputs for the air quality modelling system and other analysis.

We provided a brief explanation in the Supplementary Materials - Model Configurations and Validation in the previous manuscript. To address the reviewer's

comment, we included more explanation in the manuscript and the Supplementary Materials:

“Bulk emission inventories from the EEIO analysis were processed by the emission processing module with localized temporal and spatial surrogates to have the model-ready emission inputs for simulation and analysis (more details can be found in Supplementary Materials)”. (Line 366-368)

“The SMOKE provided model-ready emission data by allocating the annual emissions at province level into hourly interval and grid cell. Species allocations were also involved. Take Case 1 as an example, the annual bulk emission inventory for the base year of 2013 is first developed. Given that the input-output table is also in an annual basis, the bulk emission inventory is used as an input for the environmentally-extended input-output (EEIO) analysis to produce the consumption-based emission inventory. The consumption-based emission inventory reveals that how many emissions in each source category are associated with the demand of export. By excluding the emissions driven by export, a new bulk emission inventory is generated for Case 1, which is the case for us to study the impact of export. The new bulk emission inventory is processed with the temporal and spatial surrogates and emission processing systems to have the monthly, daily and even hourly emission inputs for the air quality modelling system and other analysis. The model-ready meteorological and emission data was then fed into air quality model.” (Supplementary Materials - Model Configurations and Validation)

L310-320: Are emissions from gas stations and dry cleaning included?

Response: Yes, emissions from gas stations and dry cleaning are included. In China, most gas stations adopted the way of immergence oil discharging and underground tanks, and had no control in the process of vehicle refuelling. Therefore, the NMVOCs emission factors (mg/L throughput) measured under the same situation in AP42 were selected. The oil sale was used to approximate the annual throughput in the whole China. For dry clean, a revised emission factor in the unit of kg/yr/capita from AP42 in Chapter 4 (EPA, 1995) was also selected. The emission factors per capita were calculated by multiplying conversion coefficient of 0.0246, using per-capita income as proxy variable, based on the differences in living standards between China and the US. The details of the above emissions are covered by the references mentioned in the Data Sources (Line 391-396).

Reviewer #2 (Remarks to the Author):

Review of “Role of export industries on ozone pollution in China” by Ou et al.

The paper examines the contributions of export industries and non-methane volatile organic compound (NMVOC) emissions to surface ozone pollution in China. The work is generally sound, and the writing is good, but I found the organization and explanation lacking in some key aspects.

The two topics of this paper are only very loosely related: the effects of emissions from export industries and costs and benefits of reduced NMVOC emissions. Combining them into one paper currently results in confusion. The abstract and introduction focus only on the effects of NMVOC emissions, which is the second topic. They do not mention the results of the first topic at all, which also perturbed NO_x and CO emissions as well as NMVOC emissions. As a result, it is very unclear to the reader what the processes are being tested in the simulations.

Response: We appreciate the reviewer's time and efforts on reviewing our work! The comments are critical yet very helpful for us to improve the manuscript. The comments have been addressed carefully and the current manuscript has been improved significantly.

For the first critical comment, which is about the two topics of this paper, the last section of Results and Discussion in the previous version had quite a lot of discussion on the decreasing effectiveness of NMVOCs control. We believe, that adding estimates of costs associated with proposed emission reductions is useful and assessment of cost-effectiveness of mitigation policies has been an essential part of the policy discussion and also increasingly so in China. Considering the comments of the 1st reviewer on the responsibilities of importing countries, we have revised the last section of Results and Discussion to explore how the exporting and importing countries can work together to alleviate the O₃ pollution and its precursors emissions and reduced the discussion on cost assessment. (Line 269-325)

Reviewer has questions on why we focused only on NMVOCs emissions. While studying the effects of emissions from export industries, we analysed both the NMVOCs, NO_x and CO emissions, the mitigation strategy has a strong focus on NMVOCs since the policies addressing this pollutant are lagging behind those addressing NO_x. Consequently, NMVOCs role and potential for control come more to light and we have dedicated more space to discuss those. The contributions of export to NMVOCs, NO_x and CO emissions are shown in Figure 1. In addition, Case 1 was constructed in the air quality modelling by excluding all the export-driven NMVOCs, NO_x and CO emissions. Therefore, when we studied the impact of exports on ambient O₃, we considered the changes of not only NMVOCs but also NO_x and CO. In our previous draft, however, we did not have much discussion on the NO_x and CO emissions given that some studies have covered their consumption-based emissions. To make the paper more coherent as the reviewer suggested here, we included more discussion on the NO_x and CO emissions driven by export demands in Line 128-136.

Although we did not ignore the NO_x and CO emissions from export, we do put more emphasis on NMVOCs. We explained the reasons in the Introduction:

“An understanding of the role of export industries in China’s O₃ pollution might open up new opportunities to tackle the persistent growth of O₃ and its precursors in China. In addition to the rise of ambient O₃ levels in China, its precursor - NMVOCs - is also growing persistently in contrast to the sharp decrease of NO_x and other primary pollutants [13], [14]. The persistent growth of NMVOCs is mainly due to the increase of emissions from industrial processes and solvent usage (+36%) while the NMVOCs from transport had decreased by 21% from 2010 to 2017 [14]. In addition to the contribution of O₃ formation, some NMVOCs species such as benzene, toluene, ethylbenzene and xylenes (BTEX in short) have well-documented influences on the central nervous system and immune functions of human [15]. Despite the furious discussion on a NMVOCs – or NO_x – focused control strategy to curb the tropospheric O₃ in China, NMVOCs emission control from the industry is therefore always necessary. Among China’s top export goods, many of them are associated with intensive NMVOCs emissions, including but not limited to vehicle parts, wood furniture, coke, integrated circuits, shoes and leather products. It is therefore important to understand the role of international export in China’s O₃ formation and its precursors and to explore new opportunities to curb the worrying growth of O₃ and NMVOCs in China.” (Line 71-87).

For the reviewer’s comments on the confusion of the processes being tested in the simulations, we agreed that clarifications are needed. To study the impact of export industries and the pathways to mitigate its footprint, this study sets up a few cases in the validated modeling platform: a base case and the other 3 cases (Case 1 to 3). The differences between different bases were emission inputs. To ease the confusion, we briefly mentioned the inputs of each case in the main text, followed by a more detailed description in the Supplementary. In brief, air pollutant emissions of NO_x, NMVOCs and CO for the year of 2013 were adopted in the base case, which represented the ‘true’ emissions (emissions in reality under the best knowledge) in 2013. Case 1 excluded all the export-driven emissions of NO_x, NMVOCs, and CO from the base case emissions to study the impact of export activities. Case 2 reduced 1,165 kt of NMVOCs from export-driven industrial capacities while the NO_x and CO emissions remained same as those in the base case. Case 3 reduced 4,437 kt of NMVOCs from industries by assuming that an industry-wide NMVOCs reduction effort. The NO_x and CO emissions in Case 3 remained same as those in the base case. Please see Line 158-161, 223-226, 232-238 and 329-334 in the main text for the key emission inputs for each case, followed by a detailed description in the Supplementary- Case settings.

For the analysis of export industries, it’s not clear if the emissions were changed separately for every industry in every Chinese province or if a single scale factor was applied to all sector emissions from each province.

Response: Thank you very much for raising the question. The emissions were changed separately for every industry in every Chinese province. The IO table that we were using here for China is a matrix 900*900 including 30 provinces in China and their 30 industrial sectors. Therefore, a change ratio for each sector in each province is generated for the analysis of export industries. Additional explanation is added in main text:

“The China’s MRIO table for 30 provinces and 30 sectors was linked to the GTAP database to study the impact of export and the originating countries [13], [14].” (Line 336-338)

The mortality analysis is basic. State of the art estimates of mortality from air pollution consider cause-specific mortality, rather than all-cause mortality, which varies tremendously based on the underlying reasons that people are dying. In addition, it isn’t clear if the mortality analysis assumes uniform baseline mortality across all of China, or if the mortality rates used in the calculation vary spatially.

Response: Many thanks for the suggestion. We have updated the mortality analysis to cause-specific estimation with concentration-response functions from studies within China. The increase of mortality from cardiovascular and respiratory diseases due to the exposure of O₃ is estimated based upon the concentration-response functions from Madaniyazi *et al.* (2016) [15]. When we were reviewing the evidences of epidemiological studies in China, we also found the evidence of short-term exposure of BTEX and increases in circulatory mortality [16]. Therefore, the current manuscript includes both the mortality analysis of O₃ and BTEX. More details can be found in Line 370-386 in the Materials and Methods.

Mortality analysis is conducted by gridded concentration and population in the spatial resolution of 27km× 27km. Baseline cause-specific mortality for the year of 2013 was obtained from the burden of disease study at China and the statistical yearbooks. Since mortality data is only available for in provincial level, grids within the same province adopted the provincial value. (Line 379-381)

Lesser issues

The paper uses “~” when specifying a range of values e.g. lines 22, 23, 24, , 33, 34 and many more. This is unconventional.

Response: Thanks for the comment. We used “to” or “-“ in the current manuscript.

The paper relies on Chinese government reports for background information on O₃ and PM levels, specifically references 1, 2, 6. These are in Chinese and it is not clear to me if they are peer reviewed. Link in reference 2 does not work.

Response: We thank the reviewer for the suggestion. In the previous manuscript, we cited the official reports of China's air quality from the national monitoring network (Ref 2&6) and China's air pollution control policies such as Ref 1 as the Action Plan on Air Pollution Prevention and Control. They are reliable data sources but not peer-reviewed works. Considering that they might not be accessible to a wide range of readers due to the issue of language, we changed the references to other peer-reviewed works such as the followings:

- X. Lu *et al.*, "Severe Surface Ozone Pollution in China : A Global Perspective," *Environ. Sci. Technol. Lett.*, vol. 5, no. 2, pp. 487–494, 2018. (Ref 3 in the current manuscript)
- Y. Wang *et al.*, "Contrasting trends of PM_{2.5} and surface-ozone concentrations in China from 2013 to 2017," *Natl. Sci. Rev.*, vol. 0, pp. 1–9, 2020. (Ref 12 in the current manuscript)
- B. Zheng *et al.*, "Trends in China's anthropogenic emissions since 2010 as the consequence of clean air actions," *Atmos. Chem. Phys.*, no. 18, pp. 14905–14111, 2018. (Ref 14 in the current manuscript)

References 7,8, 27 are missing journal or book titles.

Response: We are sorry for the mistakes here. The mentioned references have been corrected. And the whole reference list has been checked again.

Reference 12 is about heterogeneous chemistry, and therefore not an appropriate citation for the role of NO_x vs. NMVOC emissions ratios in controlling O₃ chemistry.

Response: Many thanks for the suggestion. Reference 12 has been replaced by the other two more relevant literatures:

- G. T. Wolff and P. E. Korsog, "Ozone control strategies based on the ratio of volatile organic compounds to nitrogen oxides," *J. Air Waste Manag. Assoc.*, vol. 42, no. 9, pp. 1173–1177, 1992.
- J. Ou *et al.*, "Ambient Ozone Control in a Photochemically Active Region: Short-Term Despiking or Long-Term Attainment?," *Environ. Sci. Technol.*, vol. 50, no. 11, pp. 5720–5728, 2016.

Please see corrections in Line 140.

Line 111: "Decrease was more notable in most O₃ hotspots in China such as the Jing-Jin-Ji, Shanxi, Guangdong and Jiangsu (Figure 2). It suggests that demand of export have slightly increased the sensitivity of O₃ formation to NMVOCs emissions ('more NMVOCs-sensitive')."

This is not convincing. A NMVOC/NO_x ratio around 1 would generally fall into the NO_x-limited regime according to most classic photochemical models (e.g. Lin and Trainer 1988; Sillman *et al.*, 1990). Satellite data also suggest that very little of China or Jing-Jin-Ji is in a VOC-limited regime, centered around urban areas (Jin and Holloway, 2015). A change from 0.94 to 0.91 in this ratio is not a meaningful change

and would not expect to be associated with the region where O₃ concentrations increase as NO_x emissions decrease

Response: We thank the reviewer for the comments. They are indeed very helpful for us to review the discussion. We completely agree with the reviewer that the decrease in the national level is not significant. The previous discussion on the emission ratio of NMVOCs and NO_x were too shallow. It is true as the reviewer suggested that China's O₃ formation are mostly governed by NO_x-limited with a few exceptions in the urban centres of Jing-Jin-Ji, Yangtze River Delta and the Pearl River Delta in Guangdong province. Given that the O₃ formation regime varies from areas, we revised the discussion on NMVOCs/NO_x emissions ratios which was on much finer spatial resolution rather than a national or provincial average.

We provided the changes of NMVOCs/NO_x emission ratios due to the demand of export in Figure 2b&c, which are the changes in every 27km*27km grid in Jing-Jin-Ji, Shandong, Yangtze River Delta, Fujian, Guangdong and Inner Mongolia. We found that the changes of NMVOCs/NO_x in local scales are much more significant than the national average. Discussion is provided in Line 138-157 in the main text. We believe the reviewer would find the analysis far more convincing now and we appreciate for the reviewer's helpful comments again.

Line 257 incomplete sentence.

Response: Sorry for the errors here. That was indeed a half sentence that should be removed.

Line 322 says "morality" instead of "mortality"

Response: Sorry for the typo. It has been corrected.

I do not have a basis for judging the reasonableness of the cost estimates in Table 1.

Response: Given that local cost information was not available, we referred to the cost of such practices in Europe using GAINS model database. With our best efforts, we provided a comparison to a study in the PRD, South China [17] in Line 261-268. And the costs for specific technologies in this study are within the range of the PRD study [17].

While we are not able to comprehensively assess the true differences between the costs in Europe and China, estimation here is generally reliable and can prove the point that cleaner production manners are affordable. For NMVOCs control, the cleaner production manners usually target the end-of-pipe treatments such as introducing activated carbon and catalytic combustion, and process management such as solvent substitution. The large proportion of the costs thus comes from the

new equipment, infrastructure, labour and the price differences between solvents. For the prices of equipment and solvents, most of them are traded internationally and the costs are quite evenly distributed between countries. For infrastructure and labour, the cost in China is even lower than that in Europe.

References

- Jin X, Holloway T. 2015. Spatial and temporal variability of ozone sensitivity over China observed from the Ozone Monitoring Instrument. *J Geophys Res* 120: 7229–7246. doi: 10.1002/2015JD023250.
- Lin X, Trainer M, Liu SC. 1988. On the nonlinearity of the tropospheric ozone production. *J Geophys Res* 93(D12): 15879–15888.
- Sillman S, Logan JA, Wofsy SC. 1990. The sensitivity of ozone to nitrogen oxides and hydrocarbons in regional ozone episodes. *J Geophys Res* 95(D2): 1837–1851. doi: 10.1029/JD095iD02p01837

References

- [1] W. Zhang *et al.*, “Unequal Exchange of Air Pollution and Economic Benefits Embodied in China’s Exports,” *Environmental Sci. Technol.*, vol. 52, pp. 3888–3898, 2018.
- [2] Q. Zhang, X. Jiang, D. Tong, S. J. Davis, H. Zhao, and G. Geng, “Transboundary health impacts of transported global air pollution and international trade,” *Nature*, vol. 543, pp. 705–709, 2017.
- [3] X. Jiang *et al.*, “Revealing the hidden health costs embodied in chinese exports,” *Environ. Sci. Technol.*, vol. 49, no. 7, pp. 4381–4388, 2015.
- [4] K. Feng *et al.*, “Outsourcing CO₂ within China,” *Proc. Natl. Acad. Sci. U. S. A.*, vol. 110, no. 28, pp. 11654–11659, 2013.
- [5] W. Zhang *et al.*, “Unequal Exchange of Air Pollution and Economic Benefits Embodied in China’s Exports,” *Environ. Sci. Technol.*, vol. 52, no. 7, pp. 3888–3898, 2018.
- [6] Economic Commission for Europe, “VOC Task Force-Emissions for Volatile Organic Compounds from Stationary Sources and Possibilities of their Control,” 1990.
- [7] M. Amann *et al.*, “Cost-effective control of air quality and greenhouse gases in Europe: Modeling and policy applications,” *Environ. Model. Softw.*, vol. 26, no. 12, pp. 1489–1501, 2011.
- [8] M. Amann *et al.*, “Progress towards the achievement of the EU’s air quality and emissions objectives,” 2018.

- [9] J. Zheng, L. Zhang, W. Che, Z. Zheng, and S. Yin, "A highly resolved temporal and spatial air pollutant emission inventory for the Pearl River Delta region, China and its uncertainty assessment," *Atmos. Environ.*, vol. 43, pp. 5112–5122, 2009.
- [10] S. Yin *et al.*, "A refined 2010-based VOC emission inventory and its improvement on modeling regional ozone in the Pearl River Delta Region, China," *Sci. Total Environ.*, vol. 514, pp. 426–438, 2015.
- [11] S. Wang, J. Zheng, F. Fu, S. Yin, and L. Zhong, "Development of an emission processing system for the Pearl River Delta Regional air quality modeling using the SMOKE model: Methodology and evaluation," *Atmos. Environ.*, vol. 45, pp. 5079–5089, 2011.
- [12] J. Zheng, W. Che, X. Wang, P. Louie, and L. Zhong, "Road-network-Based spatial allocation of on-road mobile source emissions in the Pearl River Delta region, China, and comparisons with population-based approach.," *J. Air Waste Manag. Assoc.*, vol. 59, no. 12, pp. 1405–1416, 2009.
- [13] Z. Mi, J. Meng, F. Green, D. M. Coffman, and D. Guan, "China's 'Exported Carbon' Peak: Patterns, Drivers, and Implications," *Geophys. Res. Lett.*, pp. 4309–4318, 2018.
- [14] Z. Mi, J. Meng, D. Guan, and K. Hubacek, "Chinese CO₂ emission flows have reversed since the global financial crisis," *Nat. Communications*, vol. 8, p. 1712, 2017.
- [15] L. Madaniyazi, T. Nagashima, Y. Guo, X. Pan, and S. Tong, "Projecting ozone-related mortality in East China," *Environ. Int.*, vol. 92–93, pp. 165–172, 2016.
- [16] J. Ran, H. Qiu, S. Sun, and L. Tian, "Short-term effects of ambient benzene and TEX (toluene, ethylbenzene, and xylene combined) on cardiorespiratory mortality in Hong Kong," *Environ. Int.*, vol. 117, pp. 91–98, 2018.
- [17] D. G. Streets, C. Yu, M. H. Bergin, X. Wang, and G. R. Carmichael, "Modeling study of air pollution due to the manufacture of export goods in China's Pearl River Delta," *Environ. Sci. Technol.*, vol. 40, no. 7, pp. 2099–2107, 2006.

REVIEWERS' COMMENTS:

Reviewer #1 (Remarks to the Author):

The authors have responded to reply to my comments, sincerely. It is evaluated that the responsibility of importing countries for the ozone problem in China has been added to the discussion. However, the authors should remain readers again in the discussion that Chinese domestic demand is the largest contributor to the issue and should refer to what domestic Chinese policy should be implemented. Otherwise, the readers might misunderstand the ozone issue will be solved by waiting for any countermeasures by importing countries.

Response to Reviewers' Comments

REVIEWERS' COMMENTS:

Reviewer #1 (Remarks to the Author):

The authors have responded to reply to my comments, sincerely. It is evaluated that the responsibility of importing countries for the ozone problem in China has been added to the discussion. However, the authors should remain readers again in the discussion that Chinese domestic demand is the largest contributor to the issue and should refer to what domestic Chinese policy should be implemented. Otherwise, the readers might misunderstand the ozone issue will be solved by waiting for any countermeasures by importing countries.

Response: Many thanks to Reviewer #1 for reviewing the manuscript again. We are glad that the reviewer found the comments addressed on a satisfactory level. For the remaining concern, we agree with the reviewer. Reminder is provided in Line 308-317 as:

“Even for the world’s top exporting countries like China, production for domestic market still needs to be addressed for substantial reduction of NMVOCs emissions and O₃. The direct and indirect consumption of urban and rural households in China contributed about 40% of NMVOCs emissions. With increasing household income and consumption, that contribution is expected to grow further. Policies addressing household products and consumer behaviour should be formulated. Long-term attainment of O₃ across the country would also call for further NO_x reduction of more than 50% [31]. As demand from abroad accounted for about 15% of China’s NO_x emissions in 2013, strategies targeting domestic demand driving NO_x emissions and end-of-pipe treatment would be the key to halve NO_x emission and consequently bring ambient O₃ to a safe level nation-wide.”